

# Toward insights on determining factors for high activity in antimicrobial peptides via machine learning

Hao Li and Chanin Nantasenamat

Center of Data Mining and Biomedical Informatics, Faculty of Medical Technology,
Mahidol University, Bangkok, Thailand

## ABSTRACT

The continued and general rise of antibiotic resistance in pathogenic microbes is a well-recognized global threat. Host defense peptides (HDPs), a component of the innate immune system have demonstrated promising potential to become a next generation antibiotic effective against a plethora of pathogens. While the effectiveness of antimicrobial HDPs has been extensively demonstrated in experimental studies, theoretical insights on the mechanism by which these peptides function is comparably limited. In particular, experimental studies of AMP mechanisms are limited in the number of different peptides investigated and the type of peptide parameters considered. This study makes use of the random forest algorithm for classifying the antimicrobial activity as well for identifying molecular descriptors underpinning the antimicrobial activity of investigated peptides. Subsequent manual interpretation of the identified important descriptors revealed that polarity-solubility are necessary for the membrane lytic antimicrobial activity of HDPs.

# INTRODUCTION

The continued and general rise of antibiotic resistance amongst pathogenic microbes is a well-known global threat and has been extensively reviewed before (*Hiltunen, Virta & Laine, 2017*). This threat has been cited by the United Nations to have the potential to precipitate into a global crisis (*World Health Organization, 2012*). Host defense peptides (HDPs) are defensive molecules of the innate immune system ubiquitously found amongst multi-cellular organisms (*Fjell et al., 2011*; *Evan, Straus & Hancock, 2019*). These innate immunity components are characterized by positive charge (*Torrent et al., 2011*) and amphipathicity (*Ravi et al., 2015*) and have demonstrated to be capable of directly neutralizing a vast spectrum of pathogens including bacteria, cancer, parasites, fungi, protozoa and viruses. Aside from direct pathogen neutralization, HDPs have also shown to modulate adaptive immune responses (*Hemshekhar, Anaparti & Mookherjee, 2016*).

The antimicrobial activity of HDPs has been demonstrated in many cases to be unaffected by the resistance of bacteria displayed against current antibiotics and thus has

Corresponding author
Chanin Nantasenamat,
chanin.nan@mahidol.edu

been widely suggested to be promising candidates as the next generation of antibiotics (*Li et al., 2016*). While there are no controversies in the effectiveness of HDPs against various pathogens, mechanistic understanding at the theoretical level on how HDPs neutralizes their targets is less understood. A plausible reason for this is the rather complex ways (*Mai et al., 2017*) by which HDPs interact with their targets, particularly in comparison with the often single step mechanism as displayed by classic antibiotics such as penicillin which acts as a mimicking substrate of D-alanyl-carboxypeptidase-transpeptidase (*Kelly et al., 1982*).

A significant number of dedicated HDP mechanism studies has been carried out by experimental means (*Bechinger, 2011*). It is without a doubt that experimental studies are irreplaceable as conclusive proof but they can be costly and time-consuming, particularly when few lead information are available. Furthermore, because of cost and time restrictions, experimental studies frequently has to be restricted in exploring a rather narrow space of sample and condition parameters. Hence, computational analysis of HDP mechanism can be a useful complement to experimental analysis by providing both the freedom of cost and time restriction while also providing a guide for experimental studies that would hopefully reduce the risk of fruitless experiments.

In previous articles (*Li et al., 2016*; *Shoombuatong, Schaduangrat & Nantasenamat, 2018*), we had investigated the field of HDP extensively and can attest that there is a plethora of computational studies on HDPs investigating the classification of HDPs on the basis of their target pathogen class (e.g., bacteria, cancer, parasites, etc.) (*Vishnepolsky & Pirtskhalava, 2014*; *Simeon et al., 2017*; *Beltran, Aguilera-Mendoza & Brizuela, 2018*). In addition, a large variety of HDP-related topics has been explored using computational approaches, such as computer assisted design of new HDPs (*He & He, 2016*), predicting novel HDP sequences using evolutionary computational methods (*Feng, Wang & Yu, 2017*), molecular dynamics simulations (*Petkov et al., 2018*) and 3D modeling of HDPs (*Liu et al., 2018*). Despite the variety of topics investigated, computational studies specifically exploring HDP antimicrobial strength are rare. As such, this study seeks to address this question by performing a systematic investigation on the underlying peptide parameters influencing the antimicrobial strength of HDPs.

This study applies the random forest algorithm for building a predictive quantitative structure-activity relationship (QSAR) model for modeling antimicrobial activities of HDPs. Briefly, QSAR modeling makes it possible to make sense of the large collection of existing bioactivity data by allowing the relationship that exists between the structure of compounds and their respective bioactivities to be discerned by means of statistical and machine learning approaches (*Nantasenamat et al., 2009*; *Nantasenamat, Isarankura-Na-Ayudhya & Prachayasittikul, 2010*; *Nantasenamat & Prachayasittikul, 2015*). While QSAR studies for predicting drug bioactivities are copious, this study is not merely aiming at constructing classification models of HDP antimicrobial activities but this study interprets the QSAR model on the basis of prior knowledge in the field of HDPs as to provide human understandable information on molecular parameters responsible for strong antimicrobial activities. Such insights could be readily applied for the design of effective antimicrobial peptides (AMPs).

## MATERIALS AND METHODS

### Data set

All peptide sequences, target bacteria and bioactivity data were obtained from the DBAASP database (http://dbaasp.org/home.xhtml) (*Gogoladze et al., 2014*), which represents a large collection of HDP data that had been manually curated from the open literature.

In spite of the existence of numerous online databases on HDPs, the DBAASP database appears to be the only one that provides detailed information on target organism, peptide activity and atypical residues (e.g., d-amino acids) consistently for every single peptide sequence. An empirical investigation indicated that the DBAASP database alone already provides a good approximation of the entirety of known HDPs. Particularly, Table 3 from the original report of the DBAASP database by *Gogoladze et al. (2014)* shows that the number of entries in the database at its inception ranked third in the list of major HDP databases. While the DBAASP database has fewer entries than that of the CAMP database (i.e., which is also the largest), it was explained by *Gogoladze et al. (2014)* that all entries in the DBAASP were experimentally verified whereas those from the CAMP database also contained predicted peptides (*Waghu et al., 2014*). Furthermore, a comparison was made and while in no way an exhaustive investigation, a large number of entries of one database could be found in another, not just between DBAASP and CAMP but the other databases listed in Table 3 from the article of *Gogoladze et al. (2014)* as well. Therefore, to a non trivial extent, the different databases are replicates of one another. Hence DBAASP should be a reasonable approximation of the set of all known HDPs. In addition to active antimicrobial HDPs from the DBAASP database, inactive control peptides were obtained from the UniProt (http://www.uniprot.org/) database with search conditions of "*not antimicrobial*" and "*length smaller than 60 residues.*"

### Data pre-processing

Once entries from the DBAASP database were downloaded, a series of filtering was performed on the raw data using custom scripts coded in Python or R programing languages for extracting peptide and bioactivity data that are suitable for further analysis. A single AMP can be active against multiple bacterial strains and initial screening of the raw data revealed that a very large proportion of target bacteria tested were clinical isolates or laboratory-owned samples of non-specified origin. The bioactivity of AMPs (or of any drug) depends not only on the drug but also the target; therefore, it was deemed that the AMP bioactivity measured on bacteria of non-specific origin could be of doubtful reproducibility. As this study aims to provide definitive answers that could be used to aid experimental studies, therefore it was decided to use only bacterial strains curated by the American Type Cell Collection (ATCC). Although it would be ideal to study representative microbes of clinical significance (such as ESKAPE pathogens) it was decided that due to the rather low number of peptides tested on ATCC strains, a numerical cutoff of at least 50 peptides per target strain was needed in order to retain adequate sample size for meaningful analysis. By these restrictions, seven ATCC bacterial strains were retained, these

**Table 1 Classification performance of the random forest algorithm of the AMP activities against different bacterial strains.** The numerical pIC50 values were binned to two or three levels. CV denotes 10-fold cross-validation and Test denotes test set performance. Recall performance was near perfect in all cases and not included in this table. Acc denotes the accuracy and is the proportion of peptides whose activity level has been correctly classified. Kappa stands for Cohen's kappa coefficient, which is a measure of confidence of the prediction, a value in the range of 0.4–0.6 is empirically considered moderate prediction performance (*Landis & Koch, 1977*). MCC stands for Matthew's correlation coefficient in which a value of 1 indicate perfect correlation whereas a value of 0 indicates no correlation. The supplemental file S3 (https://github.com/chaninlab/antimicrobial-peptide-QSAR/blob/master/S3.xlsx) contains all outputs relating to the prediction performance, including confusion matrices.

| Strain name | Gram | Two activity levels (CV) | | | Two activity levels (test) | | | Three activity levels (CV) | | | Three activity levels (test) | | |
|---|---|---|---|---|---|---|---|---|---|---|---|---|---|
| | | Acc | Kappa | MCC | Acc | Kappa | MCC | Acc | Kappa | MCC | Acc | Kappa | MCC |
| *B. subtilis* ATCC6633 | Pos | 0.85 ± 0.02 | 0.70 ± 0.03 | 0.71 ± 0.03 | 0.58 ± 0.00 | 0.17 ± 0.00 | 0.17 ± 0.00 | 0.79 ± 0.01 | 0.69 ± 0.02 | 0.69 ± 0.02 | 0.78 ± 0.00 | 0.78 ± 0.00 | 0.67 ± 0.00 |
| *E. faecalis* ATCC29212 | Pos | 0.64 ± 0.01 | 0.28 ± 0.03 | 0.29 ± 0.03 | 0.58 ± 0.00 | 0.17 ± 0.00 | 0.17 ± 0.00 | 0.70 ± 0.01 | 0.55 ± 0.02 | 0.55 ± 0.02 | 0.66 ± 0.02 | 0.49 ± 0.03 | 0.49 ± 0.03 |
| *S. aureus* ATCC6538 | Pos | 0.84 ± 0.01 | 0.69 ± 0.02 | 0.69 ± 0.02 | 0.63 ± 0.00 | 0.25 ± 0.00 | 0.26 ± 0.00 | 0.82 ± 0.01 | 0.73 ± 0.01 | 0.73 ± 0.01 | 0.75 ± 0.01 | 0.62 ± 0.02 | 0.63 ± 0.02 |
| *S. aureus* ATCC25923 | Pos | 0.72 ± 0.01 | 0.43 ± 0.02 | 0.44 ± 0.02 | 0.70 ± 0.02 | 0.39 ± 0.04 | 0.39 ± 0.04 | 0.81 ± 0.01 | 0.71 ± 0.01 | 0.71 ± 0.01 | 0.80 ± 0.01 | 0.70 ± 0.01 | 0.70 ± 0.01 |
| *E. coli* ATCC25726 | Neg | 0.83 ± 0.00 | 0.65 ± 0.00 | 0.66 ± 0.00 | 0.70 ± 0.00 | 0.40 ± 0.00 | 0.40 ± 0.00 | 0.85 ± 0.01 | 0.77 ± 0.02 | 0.77 ± 0.01 | 0.81 ± 0.05 | 0.71 ± 0.07 | 0.72 ± 0.07 |
| *E. coli* ATCC25922 | Neg | 0.81 ± 0.00 | 0.61 ± 0.01 | 0.62 ± 0.00 | 0.78 ± 0.01 | 0.55 ± 0.02 | 0.55 ± 0.02 | 0.84 ± 0.01 | 0.76 ± 0.01 | 0.76 ± 0.01 | 0.77 ± 0.01 | 0.66 ± 0.02 | 0.66 ± 0.02 |
| *P. aeruginosa* ATCC27853 | Neg | 0.80 ± 0.01 | 0.60 ± 0.02 | 0.60 ± 0.02 | 0.74 ± 0.02 | 0.49 ± 0.04 | 0.49 ± 0.04 | 0.84 ± 0.01 | 0.75 ± 0.01 | 0.75 ± 0.01 | 0.76 ± 0.02 | 0.64 ± 0.02 | 0.65 ± 0.02 |

are (a) *Bacillus subtilis* ATCC 6633, (b) *Enterococcus faecalis* ATCC 29212, (c) *Staphylococcus aureus* ATCC 6538, (d) *Staphylococcus aureus* ATCC 25923, (e) *Escherichia coli* ATCC 25726, (f) *Escherichia coli* ATCC 25922, (g) *Pseudomonas aeruginosa* ATCC 27853. The bacterial strain names as listed above are not in alphabetical order but in the same order as listed in Table 1 in which strains are ordered according to their Gram staining property.

Prior to analysis, peptides were subjected to several filtering steps that required peptides to possess the following characteristics: (a) must possess antimicrobial activity, (b) known to neutralize bacteria via membrane lyses, (c) contain canonical amino acids, (d) the bioactivity unit for the antimicrobial activity must be minimum inhibitory concentration (MIC), (e) the unit of the measured antimicrobial activity has to be either micromol per liter (μM) or microgram per milliliter (μg/ml), (f) the activity value must be either a scalar value or a range with upper bound (i.e., activity range given without an upper bound, such as MIC > 50 μM, were not included in this study), (g) the peptide is at least eight residues long, (h) simple terminal modifications were ignored with simple meaning modifications of the range of terminal amidation and acetylation. Peptides with complex terminal modification such as attaching fluorophores were excluded from analysis. This was done since small terminal modifications for the sake of peptide stability are unlikely to grossly alter their activity, although they do influence the peptide activity to an extent (*Park, Kim & Kim, 1998*). Furthermore, if terminal modifications are to be considered, the number of useable peptide per bacterial strain per terminal modification

type would be too small for model building as a significant number of research articles were not entirely clear on what terminal modification(s) were performed.

Conditions (d) and (e) were imposed because there were many different activity measurements and units which were not mutually convertible (e.g., MIC, $MIC_{50}$, $EC_{50}$, etc.). Furthermore, it was found that activities measured by MIC test in μM or μg/ml comprised the largest data set. AMPs whose activity was measured in μg/ml were arithmetically converted in μM and then pooled with the rest of the samples whose activity were originally published in μM. All MIC activity data were then converted into logarithmic pMIC scale as described previously (*Hevener et al., 2008*) since microbial vitality and drug concentration generally followed a logarithmic relation curve (*Hoelzer et al., 2011*; *Turnidge & Paterson, 2007*). Condition (f) was imposed because it is not possible to deduce even the approximate true activity of the peptide if the upper bound is not given, for example, a peptide with reported MIC > 50 μM could either have a true MIC of 50 μM or be completely inactive. Using this filtering criteria, the DBAASP database yielded 1,460 peptides in total with the following count for each strains: 97 *B. subtilis* ATCC 6633; 103 *Enterococcus faecalis* ATCC 29212; 128 *S. aureus* ATCC 6538; 369 *S. aureus* ATCC 25923; 84 *Escherichia coli* ATCC 25726; 423 *Escherichia coli* ATCC 25922; 256 *P. aeruginosa* ATCC 27853.

## Activity binning

Owing to the highly heterogeneous nature of the raw data (e.g., different way and different units of measuring peptide activity as stated above) it is postulated that confounding factors that may exert influence on the accuracy of the peptide activity data is the fact that the same AMPs tested in different studies can have significantly different reported activity. Also, the multiple steps of converting activity units into a unified format as well as the negligence of terminal modifications will further degrade the validity of reported AMP activity data. As such, it was deemed impractical to build an accurate numerical regression models with the available experimental data and since the primary objective of this study is to provide the reader with a readily understandable information that has promise for driving further experimental design. Therefore, it was decided that the objective of achieving model prediction accuracy would be placed secondary to the objective of interpretability. As long as the model was accurate enough to capture the gross distribution patterns of how descriptors exert their influence on activities of AMPs, it follows that useful information could be interpreted as to what descriptor patterns are required for highly active AMPs. Hence, the MIC activity data in μM was binned into three levels: high, intermediate and low activity, in which a simple three segment splits were made (https://github.com/chaninlab/antimicrobial-peptide-QSAR/blob/master/S1.xlsx contains all peptide sequence and activity data). Peptides active against a bacterial strain is first sorted by their activity values from highest to lowest, then the number of peptides is divided by three whereby the upper and lower segments forms the high and low activity level peptides, respectively. The middle segment is excluded from the analysis so as to maximize contrast between the high and low activity classes. By using this method, the exact cutoff value is dependent on the target strain/species because the activity value

range is different from target to target. *Escherichia coli* ATCC 25922, for example, has minimum and maximum MIC values of 0.059 and 338.003 µM, respectively, thereby resulting in a high/medium activity level cutoff value of 5.0 µM while a medium/low activity level cutoff value of 16.9 µM. Particularly, this means that any peptides with MIC value below 5.0 µM is considered to afford high activity while any peptides with MIC value above 16.9 µM is considered to afford low activity. By using this cutoff method, activity difference between the high and low activity levels had a difference ratio of at least 2.5 for all targets, that is, the least active peptide from the high activity level was at least 2.5 times as active as the most active peptide from the low activity level. The activity level split was performed for each individual target bacterial strain/species instead of pooling all AMPs together and splitting according to their activity. A single AMP can be active against multiple bacterial strain/species but may possess significantly different MIC values for different targets since drug activity is dependent on both drug and target properties. Different strains from the same bacterial species can have different drug sensitivity (*Xiao et al., 2005*). This study includes different bacterial species in addition to different strains, which further increases the uncertainty as to whether there is differential drug sensitivity. And while we are not aware of any study specifically discussing this topic for HDPs, the unknown influence of target physiological difference on HDP sensitivity compelled the use of definitive target strains so as to eliminate all potential influence of microbial physiology on their MIC values.

Another reason for the use of activity binning was that the raw activity data was very heterogeneous in nature, therefore peptide sequences could be reported for activity multiple times in different studies. The unit of measurement for the activity could vary from study to study. As a result, this requires arithmetic conversions so as to create a unified dataset for analysis. Moreover, discretizing the activity data into binned levels will negate the effect of small fluctuations in the raw data.

Thus, the activity binning resulted in three activity levels: high, medium and low. The important objective of this study is to give clear interpretations on what molecular descriptor patterns determine activity levels of AMPs. This was accomplished by observing the descriptor importance values as calculated by the random forest algorithm (more is given in latter text). For accurate results, the descriptor patterns of the different activity levels should be clear cut and unambiguous. Hence, the medium activity level peptides were excluded from the modeling process, minimizing the data ambiguity that the random forest algorithm has to overcome. While this approach will not give a continuous picture of the way peptide activity correlates with descriptor patterns, this approach should be able to provide an unambiguous answer as to what descriptor patterns determine high and low antimicrobial activity.

The activity binning steps resulted in a final data set consisting of 972 peptides in total, with the following count for each strain: 64 *B. subtilis* ATCC 6633; 68 *Enterococcus faecalis* ATCC 29212; 86 *S. aureus* ATCC 6538; 246 *S. aureus* ATCC 25923; 56 *Escherichia coli* ATCC 25726; 282 *Escherichia coli* ATCC 25922; 170 *P. aeruginosa* ATCC 27853. Half of the peptides for each strain belonged to the high activity level while the other half to the low activity level. In addition to active peptides, inactive peptides were included in

some of the modeling settings (more on QSAR modeling at the end of this section) in order to serve as controls.

## Molecular descriptors

Quantitative structure-activity relationship modeling essentially considers the mathematical correlation of molecular structures and their bioactivity. A prerequisite to QSAR model development is that molecules need to be described in numerical form in which the molecular structure and properties are described quantitatively or qualitatively by a set of molecular descriptors. In this study, 760 molecular descriptors suited for peptide modeling were used as follows: (a) two parameters pertaining to the molecular property namely the molecular weight (MW) and isoelectric point (PI), (b) 20 amino acid composition descriptors, (c) 400 dipeptide composition descriptors, (d) two sequence coupling number as measured by Schneider-Werder and Grantham distance, respectively, (e) 42 Quasi-sequence order descriptors as measured by Schneider-Werder and Grantham distance, respectively, (f) 42 composition, 42 transition and 210 distribution descriptors of 14 properties as given by the online Amino Acid Index database *Kawashima & Kanehisa (2000)*. Particularly, the 14 properties includes (1) hydrophobicity, (2) van der Waals volume, (3) polarity, (4) polarizability, (5) charge, (6) secondary structure, (7) solvent accessibility, (8) surface tension, (9) MW, (10) solubility in water, (11) number of hydrogen bond donor in side chain, (12) number of hydrogen bond acceptor in side chain, (13) ClogP, (14) amino acid flexibility index.

Of the descriptors class (a) MW and PI were calculated using the online EXPASY server (*Gasteiger et al., 2003*) (http://www.expasy.org). All other descriptors were calculated using the online PROFEAT server (*Rao et al., 2011*) (http://bidd2.nus.edu.sg/cgi-bin/profeat2016/main.cgi). The PROFEAT user instructions contain details of computed descriptors. Results and discussion section of this study will explain the mathematical basis and biological implications of these descriptors. It is to be noted here that since interpretation of modeled peptide activity is the main objective of this study, only descriptors for which we have a confident understanding of its mathematical principle and chemical implications will be used. Even if a descriptor results in significant prediction accuracy increase but for which we do not confidently understand its implications on the peptide activity, the descriptor will not be considered. Thus, this study places more focus on interpretability over prediction performance.

## Sequence alignment

Multiple sequence alignment guide trees (*Blackshields et al., 2010*) were calculated using the Clustal Omega (*Madeira et al., 2019*). Such guide trees were computed so as to visualize peptide sequence distances in relation to the the high and low activity class peptides (i.e., denoted as Hpep and Lpep, respectively).

## Multivariate analysis

The correlation of the AMPs property and structural descriptors to their activity levels were modeled with the random forest algorithm (*Ho, 1995*) as implemented in the Weka

data mining software (*Witten et al., 2016*). Random forest was selected as the modeling algorithm for a number of reasons as follows: (a) demonstrated robust prediction performance in wide range of domains ranging from signal processing (*Deng et al., 2017*) to social sciences (*Araque et al., 2017*), (b) relative insensitivity to initialization parameters, (c) usage familiarity by our group, (d) capable of computing molecular descriptor importance via the mean decrease of entropy (*Breiman, 2001*) (more on molecular descriptors is found at the end of this section). For a detailed description of the prediction modeling process, the book (*Kuhn & Johnson, 2013*) is suggested.

Prior to model building, the peptide data was subjected to an 80/20 ratio for stratified splitting of the initial data set by assigning 80% and 20% of the data to the training and test set, respectively. The training data was used for building the random forest model which was verified via cross-validation (using the 80% subset for both training and cross-validation) and external validation (using the 20% subset as the external test set). In construction of the QSAR model, descriptors were used as the input data matrix while the assigned activity levels for each of the AMP was used as the expected output vector. To be noted is that since HDP activity values were binned into discrete levels, the random forest algorithm is used as a classification model.

The number of descriptors used in this study was rather high (i.e., 760 altogether). As such there may be large numbers of non-informative descriptors which do not correlate with the peptide activity and would likely act as noise for the learning algorithm and thereby reduce the prediction performance. Furthermore, a large number of descriptors could drastically increase the computation time thereby rendering the repeated model building process (that was applied to compensate for statistical errors) impractical. For this study, filtering uninformative descriptors (a process known as feature selection) was performed using the *CfsSubsetEval* algorithm (*Hall, 1998*). Briefly, this algorithm is build upon the observation that informative features (descriptors) should have high correlation with the class, while having low correlation with each other. To find the absolute best set of features, exhaustive search of all combinations of feature space is the only certain way. Exhaustive search on a set of n features imposes an impossible search cost of $2^n$ possible feature subsets. To avoid this, CfsSubsetEval creates sets of features by starting from an empty set, heuristically adding new features and measuring the correlation between the selected features and the feature set with the class. If five sets of features exist whose further expansion cannot reduce either the inter-correlation between the features or the correlation of the feature set with the class in question, the algorithm is complete. CfsSubsetEval is a filter method and does not require a separate learning algorithm to run as in the case of a wrapper method. It operates on the original feature space. As such, features selected by it do not need to be interpreted in terms of a transformed feature space.

Before building the final classification model for manual analysis, initial modeling was performed by varying the tree number to 500, 1,000 and 1,500 trees. Results showed that the prediction accuracy was not of noteworthy difference for the three tested settings and as such the final model was built with 500 trees as to reduce the chance of overfitting. For building the final model that are used in the interpretation analysis, random forest modeling was performed for each of the eight bacterial strains, prediction

was repeated for 10 times and the final value was derived from the average of these runs and used for further analysis. For each of the 10 modeling repeats, the Weka random number generator was initialized with a new, physically generated high quality random seed by http://www.random.org.

Two sets of classification models were made for each of the seven strains, the first set consisted of just the active AMPs divided into two activity levels (high and low). The second set consisted of both active AMPs and the inactive control peptides from UniProt that is divided into three activity levels (high, low and inactive) in which the active AMPs are identical in the aforementioned two activity levels settings while the inactive peptides were simply appended as an additional class label. It is worthy to note that the number of inactive peptides were the same as that of the one activity level of the active AMPs (see https://github.com/chaninlab/antimicrobial-peptide-QSAR/blob/master/S1.xlsx for the peptide sequences and activity information). It is important to note that in order to maximize property contrast between the high and low activity levels and thus ease the interpretation of activity mechanism, the medium active level AMPs were deleted. The three activity level models described above serves as a control to demonstrate the ability of the RFA to differentiate between not only different activity levels of active AMPs, but also to distinguish between AMPs and random inactive peptides as well. Hence, the three activity level models were constructed using (high, low and inactive control) as opposed to (high, medium, low) activity AMPs.

In addition to Weka, a number of simple custom developed Python and R programs were used for (a) filtering the downloaded entries from the DBAASP database of peptides for further analysis, (b) formatting outputs from Weka into a suitable format for model summary. Welch's $t$-test and Kruskal–Wallis (KW) test for statistical significance was performed using built-in functions in the R programing language. While ANOVA test was perfomed using the built-in functions in Microsoft Excel.

All compiled data sets mentioned herein are provided as publicly available supplementary files on GitHub (https://github.com/chaninlab/antimicrobial-peptide-QSAR/).

# RESULTS AND DISCUSSION

## Sequence alignment

Prior to molecular descriptor calculation, there exists a possibility of using peptide sequence distances for identifying the determining factors affecting the antimicrobial activity. If the information of amino acid sequences alone were enough for the identification of activity determining factors, this would significantly simplify the predictive modeling process. Thus, this possibility was explored by visualizing peptide sequence distances by means of a multiple sequence alignment guide trees (Blackshields et al., 2010) calculated using Clustal Omega (Madeira et al., 2019). The computed guide trees (https://github.com/chaninlab/antimicrobial-peptide-QSAR/blob/master/S2.zip) shows that for all investigated bacteria types, the high and low activity class peptides (i.e., denoted as Hpep and Lpep, respectively) were significantly overlapping and could not be clearly separated. Hence, the use of molecular descriptors is not only necessary to accurately identify what physical or chemical

parameters are responsible for determining the high and low antimicrobial activity but is also necessary for the construction of accurate prediction models.

## Classification modeling

As Table 1 shows, the random forest model was able to correctly classify the activity level of HDPs. In all cases, cross-validation performances was well-behaved and showed moderate to good classification accuracy and confidence, as measured by Matthew's correlation coefficient and Cohen's kappa coefficient. Recall performance was near perfect (1.00 accuracy) in all cases and are shown in supplementary file S3 (https://github.com/chaninlab/antimicrobial-peptide-QSAR/blob/master/S3.xlsx). Results of the three activity level classification (high, low and inactive) showed that the random forest algorithm could robustly differentiate active from inactive peptides. Results from Table 1 showed that the prediction performance of the three activity level was better than that of the two activity level. This is due to the presence of the control peptides, which are readily separable from the active HDPs. It should be noted that the difference between random inactive control peptides and active HDPs were more striking than the difference between the high and low activity peptides. As such, the classification error rate for inactive control peptides is much lower than for high or low active HDPs, resulting in a higher overall accuracy for the three activity level model. Details of this can be observed from the confusion matrices in supplementary file S3 (https://github.com/chaninlab/antimicrobial-peptide-QSAR/blob/master/S3.xlsx).

## Model interpretation

The core objective of this study is to provide a readily understandable interpretation of relationship of peptide molecular descriptor patterns and antimicrobial activity. The fact that classification performance was moderate to good for all strains in both cross-validation and test set cases and for the fact that cross-validation is a valid estimate of prediction performance. Support the claim that the random forest models build in this study are able to differentiate the activity levels of the AMPs. The validity of the models indicates that there are patterns inside the molecular descriptor space by which AMPs of different activity can be distinguished. Since the random forest algorithm is capable of computing the importance value of individual descriptors, it is possible to analyze descriptor patterns correlating to peptide activity using statistical tools and human knowledge. For this purpose, only the two level classification models were analyzed to avoid confounding effect of the inactive control peptides.

Descriptors used in this study can be broadly classified into global and structural descriptors. Global descriptors are descriptors that measure properties exhibited by the entire peptide molecule such as mass, isoelectric point and amino acid composition (i.e., the percentage of a certain type of residue inside a peptide). Structural descriptors are those that take into account molecular substructure and properties (e.g., atom grids) (*Sahoo et al., 2016*). Compared to global descriptors, which is often the total sum of a given molecular property (i.e., peptide MW is the sum of the mass of all of its atoms), the calculation of structural descriptors needs to take into account the molecular structure or

property distribution topology on the molecule. As such, unlike global descriptors, which are usually single numerical values, structural descriptors are often a set of numbers whereby each describing a sub-parameter of an overall property. For example, the type of atom, bond angle and bond energy inside a crystal lattice. Thus, structural descriptors provide measurements of sub-molecular structural and property features, which are invisible on global descriptors. In the context of the present peptide study, the ability to observe molecular features below the ones exhibited by HDPs as a whole is important since their antimicrobial activities may depend on specific sites on the peptides or specific arrangements of the composing amino acids.

Of the descriptors used in this study, there are descriptors that do take sequence ordering into account, but are largely reflective of global molecular properties only. For example, dipeptide descriptors are determined by two adjacent residues, with canonical amino acids, there are 400 possible combinations. As such, dipeptide descriptors are influenced by amino acid sequence. However, a dipeptide descriptor shows only the percentage of a specific two amino acid combination in a protein chain. It is not possible to infer any meaningful sequence information about the protein chain from a dipeptide descriptor. These descriptors will be termed local structural descriptors in this study and includes: dipeptide descriptors, composition descriptors and transition descriptors. On needs to keep in mind that local structural descriptors are still essentially only describing global molecular properties.

Distribution descriptors, sequence order coupling numbers, and quasi-sequence order descriptors are significantly influenced by amino acid sequences of the entire peptide chain, as such these descriptors meet the definition of structural descriptors and will be referred to as such throughout this study. The PROFEAT instruction files contain details of the descriptors calculated by it. The mathematical basis and biological implications of the descriptors will be explained whenever they are used for analysis.

### Impact of global descriptors on peptide activity

According to previous study, global molecular properties, such as hydrophobicity and charge are the primary determinants of antimicrobial activities of HDPs and that specific sequence are not prerequisite for strong antimicrobial activities (*Oren & Shai, 1997*). An important thing to remember here is that only antimicrobial activity by membrane lyses is considered in this study.

When the important descriptors, which are defined in this study as descriptors that were deemed sufficiently informative by the *CfsSubsetEval* algorithm to be retained for the random forest classification model, were classified into global, local sequence and structural descriptors it becomes possible to assess the impact of global descriptors in the determination of peptide activity. The descriptors retained by *CfsSubsetEval* are important because these are the descriptors best suited to tell high activity AMPs apart from low activity ones.

By pooling all important descriptors from all seven strain classification together, there are 30 global, 60 local sequence (largely reflective of global properties) and 88 structural descriptors as summarized in Fig. 1. Hence, half of the important descriptors were

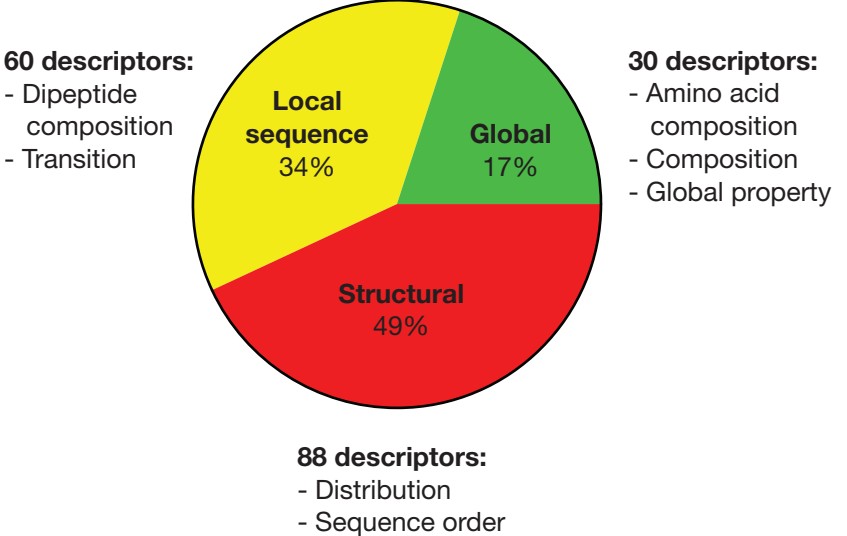

**60 descriptors:**
- Dipeptide composition
- Transition

**30 descriptors:**
- Amino acid composition
- Composition
- Global property

**88 descriptors:**
- Distribution
- Sequence order

**Figure 1  Pie chart of the pooled important descriptors of all classification instances.** The original descriptors were divided into global, local sequence and structural descriptor classes.

global property descriptors (see https://github.com/chaninlab/antimicrobial-peptide-QSAR/blob/master/S4.xlsx for the list of important descriptor names and importance value).

Aside from looking at the important descriptors of all classification instances together, Fig. 2 list important descriptor classes of individual strains by their importance value.

An empirical look at Fig. 2 indicated that global and local sequence dependent descriptors predominated in the top importance ranks for all strains investigated. That is, the most important descriptors differentiating the peptide activity levels tended to be global descriptors for all strains investigated. A more quantitative view of Fig. 2 can be obtained by dividing the list of important descriptors for each strains into four quartiles where each accounted for roughly 25% of retained important descriptors of the respective strains and calculating the proportion of descriptor classes for each quartile. As Fig. 2 shows, the uppermost quartile Q1 possessed the highest proportion of global and local sequence descriptors in all classification instances. For virtually all strains, at least 70% of Q1 were global property descriptors (it should be noted that local sequence dependent descriptors are essentially reflective only of global molecular properties and counted as global property descriptors). These observations indicated that the most important descriptors for differentiating high and low antimicrobial activity are descriptors describing molecular properties largely independent of sequence effects for all strains investigated. Hence, these observations are supportive of the view that specific sequences are not prerequisite for strong antimicrobial activities. It should be noted that Fig. 2 was obtained from the analysis of a total of 972 AMPs tested on seven bacterial strains whereas a previous study (*Oren & Shai, 1997*) experimentally tested three peptides against four bacterial strains. Hence, combining previous experimental studies with the results obtained herein, there is credible support for the notion that the antimicrobial activity of HDPs is primarily determined by global molecular properties. While there seems little

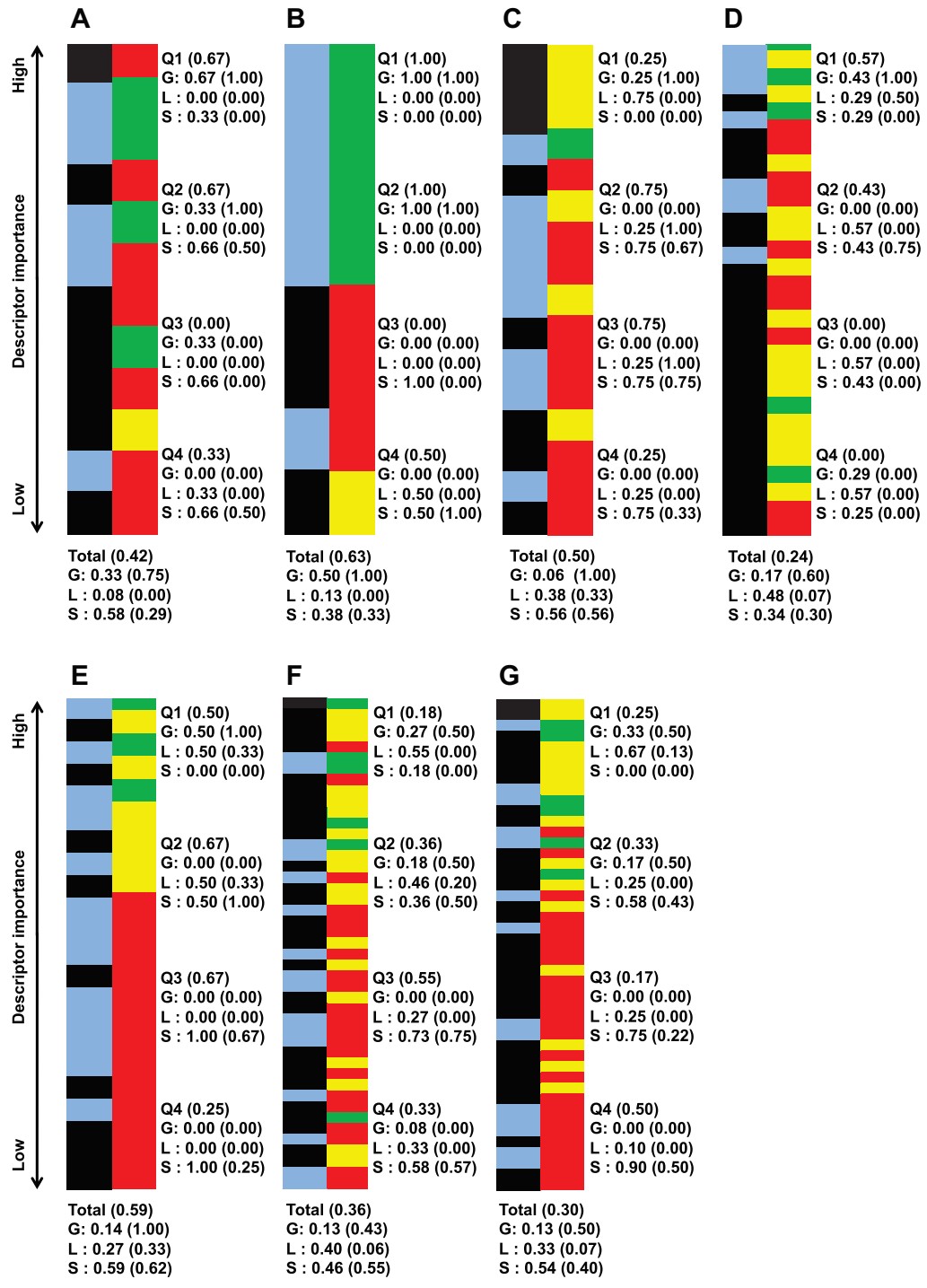

**Figure 2 Important descriptors of each target bacteria's activity level classification as ranked by their importance value (mean decrease of impurity) and classed by their degree of sequence dependence (green-red-yellow bar).** (A) *B. subtilis* ATCC 6633, (B) *E. faecalis* ATCC 29212, (C) *S. aureus* ATCC 6538, (D) *S. aureus* ATCC 25923, (E) *E. coli* ATCC 25726, (F) *E. coli* ATCC 25922 and (G) *P. aeruginosa* ATCC 27853. Green denote global descriptors, yellow denote local sequence dependent and red denote structural descriptors. The blue-black bar shows the importance distribution of descriptors pertaining to polarity-solubility (blue) and descriptors not related to polarity-solubility (black). It is worthy to note that the number of descriptors important enough to be retained by the *CfsSubsetEval* algorithm for activity

**Figure 2** (continued)
prediction is not equal for each strain. The bars of this figure representing the classes of the descriptors have been scaled to equal length for ease of comparison. An unscaled version of this figure together with raw descriptor names and importance value can be found in supplementary file S4 (https://github.com/chaninlab/antimicrobial-peptide-QSAR/blob/master/S4.xlsx). Q1–Q4 each represents 1/4 of the retained and ranked important descriptors, with Q1 containing the highest ranked and Q4 the lowest ranked descriptors. The "Total" entry at the bottom denotes the proportion of descriptor classes of all retained descriptors for each strain. Numbers in brackets stands for the proportion of descriptors related to polarity-solubility. For example, a notation of "Total (0.59)" means that 59% of all important descriptors are related to polarity-solubility. While a notation of "G (0.75)" means that 75% of global descriptors are related to polarity-solubility.

doubt that global molecular descriptors are the primary determinants of AMP activity, it must not be neglected that significant proportions of important descriptors were structural descriptors in all strains investigated, albeit possessing lower importance rank. In fact as Fig. 1 shows, a total of 50% of important descriptors were structural descriptors which are strongly sequence dependent. The study of *Oren & Shai (1997)* stated that specific sequence and peptide length were not prerequisite for strong antimicrobial activity. However, different studies (*Arias et al., 2014*; *Chen et al., 1988*; *Yan et al., 2018*) had demonstrated that the AMP sequence can significantly influence their activities. In fact, *Chen et al. (1988)* demonstrated that a single amino substitution of magainin could result in the complete inactivation of its very potent antimicrobial activity, which seems to contradict the notion that specific sequence and peptide length are not prerequisite for strong antimicrobial activity. *Oren & Shai (1997)* postulated that it could be possible that residue substitution disrupted the optimal configuration of the hydrophobicity and charge (both global properties) that already existed in magainin and hence adversely affected the peptide activity. Considering the results of previous studies and the fact that a significant proportion of important descriptors identified in this study were structural descriptors it would seem that sequence order does affect the antimicrobial activity after all, albeit not as prominently as global properties.

## Implications of the structural descriptors

The descriptor class distribution patterns of Fig. 2, could indicate the presence of strain-specific descriptor importance patterns. For the descriptor class distribution in Fig. 2 is not at all similar for each of the strains investigated, even for different strains belonging to the same species. And given the importance that structural descriptors have on the peptide activity as discussed in the previous section, it is possible that strain-specific antimicrobial activity arising from specific descriptor patterns could exist.

However, it is unlikely that the descriptor patterns in Fig. 2 is a correct representation of the strain-specific activity determining descriptor pattern, even if the assumption that the strain-specific antimicrobial activity determining descriptor pattern exists is true. Because (a) while the general antimicrobial activity mechanism of HDPs is well established, exact details as to how the peptide neutralizes the microbe is a rather complex and not nearly as clear as the mechanism of conventional antibiotics such as penicillin (*Kumar, Kizhakkedathu & Straus, 2018*) and aberrant activity mechanism cannot be ruled out. This study builds a predictive model for each strain using a large collection of different
peptides, the analysis of such classification models will only yield a gross overall view on what descriptors are important in activity determination, given the peptides used to build the classification model. Such an approach does not take into account potential mechanistic differences that may exist between the individual peptides and it is quite possible that there are sub-clusters of peptides with different sets of important descriptors for the same bacterial strain. (b) The peptide number used in this study, though large compared to experimental studies, is still not so large as to provide absolute proof of the principles of activity mechanism.

Nevertheless, it should be noticed that substantial proportion of important descriptors are strongly sequence-dependent for all strains investigated, which could be indicative of strain-specific activity mechanism. Alternatively, the sequence-dependent important descriptors could be the same amongst different strains and indicate specific sequence patterns that are required for the antimicrobial activity. Hence, it is worth to closely investigate the important descriptors in their raw format and interpret their implications toward antimicrobial activities to the fullest extent as permitted by the available results and knowledge. The following sections will explain that the results do not support strain-specific antimicrobial mechanism, rather the AMPs antimicrobial activity builds upon a general polarity-solubility dependent mechanism independent of the target bacterial strain.

## Overview of the raw important descriptors

When all important descriptors that form Fig. 2 were pooled together in the raw format and disregarding their importance value, there exists a total of 138 unique descriptors with 178 total occurrences. Occurrence means the times a descriptor has been retained as important by the *CfsSubsetEval* algorithm. Initially, all classification models were built with the same 760 descriptors. Each descriptor represents a unique named descriptor (e.g., MW) that can be retained as important in more than one strain classification model (e.g., in the model of *Escherichia coli* ATCC 25922 and *Escherichia coli* ATCC 25726). As such, the unique descriptor MW has an occurrence of 2 as it was found to be important for two bacterial strains.

If multiple strains shared an important descriptor, it would indicate that descriptor represents a common molecular parameter by which high and low antimicrobial activity separates. It was found that 138 unique descriptors account for 77% of the total of 178 occurrences of the important descriptors thereby indicating low overlap of the important descriptors that determine the activity level of AMPs targeted to different bacterial strains.

At a glance, this seems to support the idea of strain-specific activity mechanism. If different target bacteria depend on very similar descriptors for their activity, the list of important descriptors of the different target strains should look very similar and the number of unique descriptors should be low. However, detailed investigation shows that while the different strains do have different sets of important descriptors, most of the important descriptors are descriptions of peptide polarity and solubility.
## Global descriptor details

To further elucidate how the descriptors affected the antimicrobial activity, an analysis of the important descriptors in their raw form for the individual strains was performed. When the important global descriptors of all strains were pooled together, there were 19 unique descriptors (https://github.com/chaninlab/antimicrobial-peptide-QSAR/blob/master/S5.xlsx) with a total occurrence count of 30. At first glance, 19 unique descriptors with 30 occurrences indicated that the different strain classification do not share a lot of important descriptors. However, closer observation revealed that the important global descriptors of each strain are closely related and in fact belonged to a very narrow category of property parameters.

Composition descriptors constituted the vast majority of important global descriptors. Composition descriptors expressed the percentage of amino acid property classes present in a peptide. It is calculated by firstly dividing amino acids by a property (e.g., charge) into three classes (positive, neutral, negative) and dividing the number of residues in each class by the total number of residues constituting a peptide. For example, a 10 residue peptide consisting of one positive and nine negative residues gives the following composition descriptors (a) Composition.of.Charge.1 = 0.1 (b) Composition.of.Charge.2 = 0.0 (c) Composition.of.Charge.3 = 0.9. The descriptor naming is the raw output of PROFEAT, with Charge 1, 2, 3 denoting class of positive, neutral and negative amino acids respectively. The PROFEAT manual found on the website contained details on the calculation of all descriptors. The descriptor range for dividing properties classes as done by PROFEAT, which was originally developed by *Dubchak et al. (1995)*. A simplified version can be found in the PROFEAT manual.

The important composition descriptors were parameters for charge, ClogP, hydrophobicity, side chain hydrogen bond donor, van der Waals volume, polarity, polarizability, solvent accessibility, surface tension and secondary structure. With the exception to secondary structure, all the composition descriptors were parameters closely related to peptide charge and solubility in a vice versa manner. In addition, isoelectric point too is closely related to peptide charge and in extension to solubility as well. And as Fig. 2 shows, a large percentage of global descriptors are polarity-solubility related. As such, there is little doubt that charge and solubility are the most important global molecular parameters for separating high and low antimicrobial activity of AMPs. The observation made in this study is consistent with previous experimental results of *Oren & Shai (1997)*, which stipulates that hydrophobicity and charge are the primary determinants of antimicrobial activities of AMPs. The same observation was made by a number of research and review articles (*Čeřovský et al., 2008*; *Li et al., 2016*; *Toro Segovia et al., 2017*) where all of which made the observation that positive charge and amphipahicity was important for AMP antimicrobial activity. In fact, positive charge is a key recognition feature of AMPs and antimicrobial activity could be significantly increased by simply adding more positively charged amino acids to the peptides (*Papo & Shai, 2003*).

As Fig. 2 shows, high percentage of important global descriptors pertained to polarity-solubility for all target bacterial strain/species. The strains investigated in this
study include both Gram-positive and Gram-negative bacteria (Table 1) and considering the multiple different strains investigated. Results from this study lends further support to the theory that AMP bacterial membrane lyses rely on a common mechanism that involves the binding of positively-charged HDPs to the negatively-charged bacterial membrane and disrupting the membrane integrity.

In addition to analyzing important global descriptors for all strains together, a detailed analysis of the important global descriptors of each individual strain was also carried out to give a clear answer on how they affect the peptide activity against specific bacterial strains. An issue in discussing strain-specific raw descriptor importance is that each strain possesses a different list of retained important descriptor and has different importance values for every descriptor as well. While it is possible to discuss how the activity levels of each individual strain differ by their respective important descriptors, it would not give a good comparison on how global descriptors affect peptide activity against different strains and as previously discussed, there is reason to believe that activity against different bacterial strains are determined by a common polarity-solubility property complex. A further problem of discussing each strain separately is that it would result in an excessively long discussion which cannot be fit into a single publication.

In order to facilitate comparison between the strains, discussion of the important global descriptor was carried out by focusing on descriptor distribution patterns observable amongst different strains rather than discussing each strain separately. A problem with such a method is that it would result in the treatment of a descriptor as important for all strains even though it was retained as important in the classification model of just one strain only. However, by analyzing the importance value of the retained descriptors and relating them to the known activity mechanism of AMPs, it is possible to identify what descriptor patterns are related to the known activity mechanism of AMPs. While those descriptors that were retained as important but cannot be explained via a known HDP activity mechanism are likely indicative of either strain-specific activity mechanism or unknown common activity mechanism. And as the analysis result shows, the global descriptor patterns agree very well with what is known about the HDP antimicrobial activity mechanism and does not indicate strain-specific activity mechanism.

## Trends in the important global descriptors

A remarkable consistency can be observed when the raw descriptor values of the retained global descriptors were averaged (Table 2). All global descriptors retained as important for activity classification in more than one strains had identical trends in their averaged value. To illustrate, Composition.of.CLogP.2 was retained as an important descriptor for three different strains, *Enterococcus faecalis* 29212, *S. aureus* 6538 and *S. aureus* 25923. The averaged value of that descriptor was lower for the high activity level peptides as compared to the low activity level peptides for all three strains. In addition to showing the average descriptor values for the high and low activity level HDPs, Table 2 also list the results from the Welch's *t*-test of significance, which shows whether the null hypothesis has been rejected and if there is significant difference between the average descriptor values of high and low activity HDPs. As summarized in Table 2, it was shown that the
**Table 2 Summary of averaged descriptor value and standard deviations of all important global and non-dipeptide local sequence descriptors for various strains.** Relative differences in the averaged value of important descriptors between high and low activity AMPs can provide information about the activity mechanism. For each bacterial species, four pieces of information are shown (from top to bottom) as follows: (1) averaged value and standard deviation of the descriptor from the high activity class, (2) averaged value and standard deviation of the descriptor from the low activity class, (3) $p$-value from the Welch's $t$-test, (4) whether the null hypothesis (no significant difference between high and low activity class of the descriptor's mean) was rejected. It should be noted that the rejection threshold used was $p$-value < 0.05.

| Descriptors | B. subtilis ATCC6633 | E. faecalis ATCC29212 | S. aureus ATCC6538 | S. aureus ATCC25923 | E. coli ATCC25726 | E. coli ATCC25922 | P. aeruginosa ATCC27853 |
|---|---|---|---|---|---|---|---|
| Important global descriptors | | | | | | | |
| C | – | – | – | 0.00 ± 0.00 | – | 0.06 ± 0.55 | – |
| | – | – | – | 0.43 ± 1.94 | – | 0.51 ± 2.09 | – |
| | – | – | – | 1.40E−02 | – | 1.62E−02 | – |
| | – | – | – | Yes | – | Yes | – |
| Composition.of. Charge.1 | 32.57 ± 10.27 | – | – | 29.65 ± 15.15 | – | – | – |
| | 20.75 ± 13.21 | – | – | 22.98 ± 12.10 | – | – | – |
| | 1.80E−04 | – | – | 1.77E−04 | – | – | – |
| | Yes | – | – | Yes | – | – | – |
| Composition.of. Charge.2 | – | – | – | – | 75.73 ± 6.66 | – | – |
| | – | – | – | – | 83.74 ± 6.00 | – | – |
| | – | – | – | – | 1.74E−05 | – | – |
| | – | – | – | – | Yes | – | – |
| Composition.of. CLogP.2 | – | 27.58 ± 12.94 | 14.16 ± 13.32 | 29.25 ± 16.19 | – | – | – |
| | – | 31.52 ± 16.04 | 29.32 ± 23.11 | 37.26 ± 13.01 | – | – | – |
| | – | 2.67E−01 | 4.00E−04 | 2.79E−05 | – | – | – |
| | – | No | Yes | Yes | – | – | – |
| Composition.of. No.of.hydrogen. bond.donor.in. side.chain.1 | 34.33 ± 9.00 | 35.47 ± 9.39 | – | – | – | 33.48 ± 12.61 | – |
| | 27.83 ± 15.11 | 28.50 ± 14.05 | – | – | – | 29.16 ± 14.27 | – |
| | 4.17E−02 | 1.92E−02 | – | – | – | 7.39E−03 | – |
| | Yes | Yes | – | – | – | Yes | – |
| Composition.of. No.of.hydrogen. bond.donor.in. side.chain.2 | – | – | – | – | 10.49 ± 3.41 | 11.75 ± 7.09 | – |
| | – | – | – | – | 13.69 ± 6.05 | 15.04 ± 1 0.33 | – |
| | – | – | – | – | 1.91E−02 | 6.29E−04 | – |
| | – | – | – | – | Yes | Yes | – |
| Composition.of. No.of.hydrogen. bond.donor.in. side.chain.3 | – | 52.16 ± 14.79 | – | – | – | – | – |
| | – | 57.50 ± 23.41 | – | – | – | – | – |
| | – | 2.66E−01 | – | – | – | – | – |
| | – | No | – | – | – | – | – |
| Composition.of. Normalized. vdW.volumes.3 | – | – | – | – | – | – | 47.16 ± 16.34 |
| | – | – | – | – | – | – | 40.34 ± 22.37 |
| | – | – | – | – | – | – | 2.48E−02 |
| | – | – | – | – | – | – | Yes |
| Composition.of. Polarity.2 | – | – | – | 21.31 ± 14.67 | – | – | – |
| | – | – | – | 28.70 ± 13.96 | – | – | – |
| | – | – | – | 6.92E−05 | – | – | – |
| | – | – | – | Yes | – | – | – |

| Descriptors | B. subtilis ATCC6633 | E. faecalis ATCC29212 | S. aureus ATCC6538 | S. aureus ATCC25923 | E. coli ATCC25726 | E. coli ATCC25922 | P. aeruginosa ATCC27853 |
|---|---|---|---|---|---|---|---|
| **Table 2 (continued).** | | | | | | | |
| Composition.of. Polarizability.1 | – | 13.38 ± 12.44 | – | – | – | – | – |
| | – | 19.81 ± 14.80 | – | – | – | – | – |
| | – | 5.70E−02 | – | – | – | – | – |
| | – | No | – | – | – | – | – |
| Composition.of. Polarizability.3 | – | – | – | – | – | – | 47.16 ± 16.34 |
| | – | – | – | – | – | – | 40.34 ± 22.38 |
| | – | – | – | – | – | – | 2.48E−02 |
| | – | – | – | – | – | – | Yes |
| Composition.of. Secondary. structure.2 | 19.90 ± 12.34 | – | – | – | – | 23.17 ± 9.56 | – |
| | 26.50 ± 8.37 | – | – | – | – | 27.00 ± 11.70 | |
| | 1.54E−02 | – | – | – | – | 2.86E−03 | – |
| | Yes | – | – | – | – | Yes | |
| Composition.of. Solvent. accessibility.1s | – | – | – | – | – | – | 49.63 ± 10.58 |
| | – | – | – | – | – | – | 58.35 ± 13.33 |
| | – | – | – | – | – | – | 5.00E−06 |
| | – | – | – | – | – | – | Yes |
| Composition.of. Surface. tension.3 | – | – | – | – | – | 42.33 ± 13.97 | – |
| | – | – | – | – | – | 46.60 ± 13.31 | |
| | – | – | – | – | – | 8.98E−03 | – |
| | – | – | – | – | – | Yes | – |
| Isoelectric | 10.68 ± 0.72 | – | – | – | 10.39 ± 0.29 | – | 11.25 ± 1.04 |
| | 9.57 ± 1.27 | – | – | – | 9.81 ± 0.71 | – | 10.34 ± 1.18 |
| | 7.98E−05 | – | – | – | 2.80E−04 | – | 3.29E−07 |
| | Yes | – | – | – | yes | – | Yes |
| M | – | – | – | 0.48 ± 1.34 | – | – | – |
| | – | – | – | 1.19 ± 2.51 | – | – | – |
| | – | – | – | 6.29E−03 | – | – | – |
| | – | – | – | Yes | – | – | – |
| Mass | – | – | – | – | – | 2,752.86 ± 1,162.02 | 2,960.49 ± 1,086.25 |
| | – | – | – | – | – | 1,855.92 ± 672.30 | 1,926.44 ± 729.44 |
| | – | – | – | – | – | 1.00E−13 | 1.81E−11 |
| | – | – | – | – | – | Yes | Yes |
| P | – | – | – | – | – | – | 2.78 ± 3.69 |
| | – | – | – | – | – | – | 1.04 ± 2.61 |
| | – | – | – | – | – | – | 5.39E−04 |
| | – | – | – | – | – | – | Yes |
| S | – | – | – | – | – | 3.46 ± 4.09 | – |
| | – | – | – | – | – | 5.79 ± 6.16 | – |
| | – | – | – | – | – | 9.80E−04 | – |
| | – | – | – | – | – | Yes | – |

(Continued)

| Descriptors | B. subtilis ATCC6633 | E. faecalis ATCC29212 | S. aureus ATCC6538 | S. aureus ATCC25923 | E. coli ATCC25726 | E. coli ATCC25922 | P. aeruginosa ATCC27853 |
|---|---|---|---|---|---|---|---|
| Important local sequence descriptors | | | | | | | |
| Transition.of. Charge.3 | – | – | – | – | 2.56 ± 2.88 | – | – |
| | – | – | – | – | 3.99 ± 4.91 | – | – |
| | – | – | – | – | 1.90E−01 | – | – |
| | – | – | – | – | No | – | – |
| Transition.of.No. of.hydrogen. bond.donor.in. side.chain.3 | – | – | – | – | 11.34 ±4.16 | 12.90 ± 8.01 | 11.91 ± 7.65 |
| | – | – | – | – | 15.93 ± 6.21 | 17.28 ± 10.10 | 16.61 ± 8.52 |
| | – | – | – | – | 2.16E−03 | 7.57E−05 | 2.14E−04 |
| | – | – | – | – | Yes | Yes | Yes |
| Transition.of. Normalized. vdW.volumes.1 | – | – | 8.90 ± 14.02 | – | – | – | – |
| | – | – | 19.59 ± 19.55 | – | – | – | – |
| | – | – | 4.70E−03 | – | – | – | – |
| | – | – | Yes | – | – | – | – |
| Transition.of. Secondary. structure.1 | – | – | – | – | 30.51 ± 9.72 | – | – |
| | – | – | – | – | 22.63 ± 8.43 | – | – |
| | – | – | – | – | 2.07E−03 | – | – |
| | – | – | – | – | Yes | – | – |
| Transition.of. Solvent. accessibility.3 | – | – | – | 5.94 ± 5.76 | – | – | – |
| | – | – | – | 6.88 ± 7.71 | – | – | – |
| | – | – | – | 2.81E−01 | – | – | – |
| | – | – | – | No | – | – | – |

average differences between high and low activity level HDPs were significant (i.e., with few exceptions). And while there are descriptors whose difference were not high enough to reject the null hypothesis, their impact must not be neglected either because ensemble classification methods such as random forest do not make decisions based on a single factor. Hence, while not all descriptors possessed sufficient difference for statistical significance, whether these or indeed any descriptor actually influence the peptide activity will need to be assessed holistically on the basis of knowledge pertaining to the peptide antimicrobial mechanism. As such, such analysis will be carried out in detail in this section.

This section will analyze the influence of global descriptors on peptide activity as centered on the trends of the averaged values discussed above. As Table 2 shows, different strains retained different global descriptors as important ones thus making analysis of their influence on the antimicrobial activity problematic. However, as discussed previously, nearly all of the retained global descriptors are closely related to charge-solubility and all descriptors had identical trends in their average values for different strains. This observation supports that global molecular descriptors influence antimicrobial activity via a common mechanism that is independent of the bacterial strains and Gram property. It is therefore possible to discuss influence of different descriptors by relating them to the charge-solubility framework. To accomplish this, an interpretation
of the important global descriptors will be first be given and then related to their influence on molecular charge-solubility.

Composition.of.Charge.1 and 2 (CoC1 and CoC2) descriptors account for the percentage of positive and neutral residues respectively inside a peptide. These two descriptors were retained as important in the activity modeling three different strains (Table 2). Average values of CoC1 were higher for highly active peptides while the average values of CoC2 were lower for highly active peptides. Thus, these values indicate that HDPs with high antimicrobial activity possessed more positively charged residues on average.

Composition.of.CLogP.2 (CLogP2) describes the percentage of intermediate soluble residues. A compound with high logP value has low solubility in water. ClogP stands for LogP value adjusted for molecular fragment contribution. All strains which retained CLogP2 as an important descriptor had lower average value for peptides of high activity level.

Composition.of.No.of.hydrogen.bond.donor.in.side.chain.1, 2 and 3 (Chbdo1, Chbdo2, Chbdo3) descriptors count the percentage of residues with more than one hydrogen bond donor (Chbdo1), exactly one hydrogen bond donor (Chbdo2) and no donor (Chbdo3). It was calculated that Chbdo1 was higher for high active HDPs while Chbdo2 and Chbdo3 was lower for high active HDPs.

Composition.of.Normalized vdW volumes.3 (CVdW3) is the percentage of residues with high van der Waals volume (4.03–8.08). The van der Waals volume of an amino acid is calculated from the collective van der Waals radius of its constituent atoms. A residue with high van der Waals volume more readily forms intermolecular van der Waals bonds via weak London dispersion force and stronger dipole-dipole force. In the context of amino acids, the strong dipole force means greater solubility in water. It was found the high activity peptides had higher CVdW3.

Composition.of.Polarity.2 is the percentage of residues intermediate polarity, defined in PROFEAT as the amino acid P, A, T, G, S with polarity index between 8.0 and 9.2. The high active peptides had lower percentage of these residues. Being of intermediate class, its activity implication is problematic to analyze, additionally this descriptor was retained only once in the model of *S. auraus* 25923 and has no related retained descriptors of the same class. Hence, it was not deemed informative enough and was excluded from further analysis.

Composition.of.Polarizability.1, 2 and 3 (CoPl1, 2 and 3) are the percentages of residues with low (<1.08), intermediate (0.12–0.18) and high (0.22–0.41) degree of polarizability. It was found that high activity peptides possessed a greater percentage of high polarizable residues (CoPl3) while having lower percentage of low and intermediate polarizable residues.

Composition.of.Secondary.structure.2 (CoSs2) is the percentage of strand-forming residues. It was found that high activity peptides possessed lower percentage of them. To give a better perspective of the influence of secondary structure, the values for percentage of helix (CoSs1) and coil (CoSs3) forming residues were also calculated. It was found that in all models where CoSs2 was retained, high activity peptides had a higher percentage of helix forming residues and a lower percentage of strand and coil forming residues. It has been experimentally shown that high helicity is positively correlated with

antimicrobial activity (*Chen et al., 2005*), it is therefore not surprising that high activity peptides had higher CoSs1. However, that CoSs2 was retained as important descriptor is unexpected. It may be due to the fact that all peptides considered in the two activity level models were AMPs and already possessed a high content of helix forming residues and hence making it less ideal for fine grained activity differentiation. As a result, strand forming residue differences gave better resolution of peptide activity levels.

Composition.of.Solvent.accessibility.1 (CoSac1) is the percentage of residues that tend to be buried in protein backbone and not solvent exposed. Usually these are hydrophobic residues such as tryptophan. High activity peptides were found to have a lower percentage of buried residues.

Composition.of.Surface.tension.3 (CoSut3) is the percentage of high surface tension residues and it was observed that high activity peptides had on average lower percentage of residues with high surface tension. Surface tension is a measure of energy cost of increase of surface between two phases. If a molecule is only surrounded by the same kind of molecules, this energy is minimized, whereas when coming into contact with another kind of molecule, an energy barrier needs to be overcome to create an interface. Surface tension is a measure of this energy barrier to be overcome. The higher the surface tension, the harder it is to create more interface between two phase hence the two phase do not mix easily. A surface tension of zero means the absence of an interface barrier and the two phases are fully miscible. The surface tension of this study is a measure of the miscibility of amino acids and water, therefore CoSut3 can be seen as the percentage of low soluble amino acids. Highly active peptide had a lower percentage of low soluble residues.

Isoelectric point is a global descriptor retained as important by four different strain models, in all cases, highly active peptides had on average higher isoelectric point. It is also noteworthy that this descriptor is a parameter of the peptide as a whole, rather than any component of it. At pH below their PI, proteins carry net positive charge. The observation that high active peptides tend to have higher PI indicate that high activity peptides retain their positive charge over a greater pH range than low activity ones.

Molecular mass was higher for high activity AMPs. This study is unable to deduce the biological implication of a higher mass. Similarly, the amino acids composition (C, M, P) descriptors retained as important can be presented as observed results only.

Observing the distribution patterns of the average values of high and low activity HDPs against different bacterial strains as described above, it can be seen that even though none of the important descriptors were retained for all strain classification. Their distribution does not contradict what is known about factors increasing AMP antimicrobial activity, namely high positive charge and amphipathicity (*Torrent et al., 2011*). High positive charge is observed in this study in the form of descriptors representing higher percentage of positively charged residue and a greater polarizability potential. Stable coil is indicated by higher percentage of helix forming residues. While descriptors used in this study are not directly related to amphipathicity, high helical content is associated with high amphipathicity. Also, the seemingly contradictory observations made of descriptor patterns indicating high solubility and high hydrophobicity at the same time could be an indication of high amphipathicity. The contradictory descriptor patterns are: high
polarizability, greater number of hydrogen bond donors, lower percentage of buried residues, lower percentage of low soluble residues, higher van der Waals volume. These descriptor patterns all indicate high solubility, yet it was also observed that high activity peptides had high percentage of hydrophobic residues. Hence, summarizing all analysis for global descriptors together, it is observable that global molecular parameters of charge and solubility are the primary determinants of AMP antimicrobial activity. All observations made in this study regarding the global descriptors are at least not contradictory with previous results suggesting strong positive charge and amphipathicity are the main factors for strong antimicrobial activity. The consistency of results obtained by this computational study with previous experimental results is a good support for the validity of the computational models created in this study.

In all, no evidence for strain-specific distribution was observed for the global descriptor, indicating that global molecular properties influencing antimicrobial activity underlies a general action mechanism. While the number of strains investigated in this study is not high enough to definitely conclude this strain independence. Given the results of published literature which in general indicates an independence of antimicrobial activity from bacteria strain and drug resistance and the results obtained in this study. It can be said with good confidence that the main antimicrobial activities of HDPs as determined by charge-solubility related parameters are not significantly specific to particular bacterial strains or Gram property. The results of this section also demonstrates that the analysis methods used in discussing retained important descriptor of different strains by first assuming they influence a common activity mechanism and then relating the descriptor value patterns to known activity mechanism can be used to deal with the issue of each strain possessing a different set of retained important descriptors.

## Local sequence order descriptor details

In a similar vein as that of global descriptors, local sequence order descriptors were analyzed for their influence on the peptide activity. As shown in supplementary file S5 (https://github.com/chaninlab/antimicrobial-peptide-QSAR/blob/master/S5.xlsx), 50 local sequence order descriptors were retained for 60 times in seven activity classification models. Furthermore, 43 out of the 50 (86%) local sequence descriptors were dipeptide descriptors while another seven were transition descriptors. Briefly, transition descriptors were calculated by dividing amino acids according to various properties in the same fashion as composition descriptors discussed in the previous section. For example, amino acids were divided according to their solubility into polar, neutral and hydrophobic classes represented by class index 1, 2, 3 respectively. A transition occurs if two adjacent residues are of different classes, for example, a neutral residue followed by a hydrophobic one gives the transition "13." By transforming a peptide sequence into their property class indexes and calculating the percentage of each type of transition permutation one obtains the transition descriptors.

A look at the transition descriptors (https://github.com/chaninlab/antimicrobial-peptide-QSAR/blob/master/S5.xlsx) shows that they are reflective of the same properties, namely polarity-solubility, as the composition descriptors discussed earlier. This is to

be expected as local sequence order descriptors are still largely reflective of global molecular properties.

Similar to the composition descriptors, analysis of the transition descriptors is complicated by the fact that few of them are shared amongst the different strain classification models. However, as they are all about polarity-solubility it is possible to relate the value distribution patterns of the different transition descriptor to known HDP mechanism in the same way as the composition descriptors discussed in the previous section. The values of the transition descriptors can be found in Table 2 and at GitHub (https://github.com/chaninlab/antimicrobial-peptide-QSAR/blob/master/S1.xlsx).

Transition.of.CLogP.1 and 2 (ToClogP1 and 2) are the percentage of (a) ToClogP1: a hydrophilic residue followed by an intermediate soluble one ("12" residue class index) and (b) ToClogP2: a hydrophilic followed by a hydrophobic residue ("13" residue class index). It was found that high activity peptides had lower ToClogP1 but higher ToClogP2. Indicating at antimicrobial activity could be associated with inflection of solubility along the peptide chain. This solubility inflection maybe an indication of the structural basis of the amphipathicity required for antimicrobial activity.

Transition.of.Normalized.vdW.volumes.1 (TovdW1) is the percentage of one low van der Waals volume residue followed by another residue with intermediate van der Waals volume ("12" residue class index), see the PROFEAT manual for detailed description for classifying van der Waals volume. It was found that high activity peptides had on average only half the value of TovdW1 compared to low activity peptides. As discussed for the global descriptors, in the context of amino acids, high van der Waals volume imply greater solubility in water. A low content of low vdW residue followed by another intermediate vdW volume can imply the preference for having few chain segments with low van der Waals bond potential. However this descriptor was retained by only one strain model and a lower content of chain segements with low van der Waals bond potential by itself is not relatable to any known peptide mechanism but nor does it contradict any know mechanism. Hence, it was deemed this descriptor was not informative enough to either support or reject strain-specific activity mechanism, more data would be needed for a definitive answer.

Besides transition descriptor, dipeptide descriptors were important local sequence order descriptors, composing the vast majority of retained important local sequence order descriptors. It was found that very few dipeptide descriptors were retained as important in more than one strain model. In fact, there was no dipeptide that was retained as important in more than two strain models.

No obvious patterns could be discerned in the list of retained dipeptide descriptors. This large diversity of important descriptors of local structure is likely indicative of the diversity of local peptide structures capable of creating the necessary polarity-solubility configuration of the peptide for antimicrobial activity.

## Structural descriptor details

Descriptors that are strongly sequence order dependent are structural descriptors. These are parameters that reflect molecular sub structures and properties that are carried by such substructures (*Sahoo et al., 2016*). As seen in supplementary file S5 (https://github.

com/chaninlab/antimicrobial-peptide-QSAR/blob/master/S5.xlsx), there exists a high diversity in the important structural descriptors from the different strains. As a result, a detailed discussion on the influence of this class of descriptors on the activity of HDPs against each different bacterial strain would necessitate a lengthy explanation.

However, a close look supplementary file S5 (https://github.com/chaninlab/ antimicrobial-peptide-QSAR/blob/master/S5.xlsx) reveals that while the retained structural descriptors were diverse, the vast majority of these descriptors are once again reflective of the polarity-solubility property complex as identified by analysis of both global and local sequence dependent descriptors (https://github.com/chaninlab/antimicrobial-peptide-QSAR/). Of the 69 unique structural descriptors retained, 57 (83%) were distribution descriptors of the properties of (a) amino acid flexibility, (b) charge, (c) ClogP, (d) hydrophobicity, (e) MW, (f) number of hydrogen bond acceptor in side chain, (g) number of hydrogen bond donor in side chain, (h) normalized van der Waals volume, (i) polarity, (j) polarizablity, (k) Secondary structure, (l) solublity, (m) solvent accessibility and (n) surface tension. It should be noted that all of these descriptors are inherently related to the polarity-solubility property complex.

As mentioned at the section start, retained distribution descriptors were very diverse and no easily distinguishable pattern could be observed. Hence, results of this study is insufficient to definitely conclude whether the greatly varying patterns of the distribution descriptors as well as that of other retained structural descriptors reflect strain-specificity or not. However, note that polarity-solubility related descriptor were consistently important throughout the class of global, local and structural descriptors and occupied an overwhelming proportion of the retained global descriptors. Hence, polarity-solubility complex is an antimicrobial activity determining factor that remains important throughout global, local and structural level of HDPs.

Nevertheless, while an overall shared antimicrobial mechanism is indicated by all available evidences, the difference of retained important descriptors amongst the different strain could imply non-negligible difference of the HDPs to achieve optimal neutralization of a particular strain. It is worth to note here that HDP membrane lyses can occur in different ways including barrel stave, carpet and torrodial pores (*Li et al., 2016*) models. All three pathways are intimately related to peptide charge and amphipathicity but do have non-trivial differences.

## Distribution of importance values of polarity-solubility descriptors

The above discussions in conjunction with results from previous studies indicates the general dependence of antimicrobial activity on descriptors pertaining to polarity-solubility related parameters in all target strain/species investigated. In order to investigate whether the dependence of activity on polarity-solubility is the same amongst different targets, the important descriptors for each target was classified into either polarity-solubility descriptor or not as represented by blue and black, respectively, in Fig. 2. At a glance, the different target seems to have different importance distribution patterns for their polarity-solubility related descriptors. To gain a more objective measurement, statistical test of variance with both ANOVA and KW test were carried out for the

proportion values of all polarity-solubility descriptors. Both tests failed to reject the Null hypothesis with ANOVA $F$-value = 0.94, $F$-critical = 2.45, KW-test $p$-value = 0.31 and $p$-threshold = 0.05. Hence, whether or not the importance distribution of the polarity-solubility descriptors is assumed to be normal, the results indicates that the distribution of the polarity-solubility related descriptors are not statistically different for the different target microbes. Therefore, the results supports the observation that antimicrobial activity strength of HDPs depends on a similar pattern of polarity-solubility related descriptors independent of the target bacteria in question.

## General requirements for high activity

Table 2 contains a detailed listing of important descriptors and their distribution values for high and low activity HDPs. In addition to providing the data for analyzing factors influencing the HDP activity, information from Table 2 can also be used in assisting the synthesis of novel HDPs. For example, high antimicrobial activity could be expected from a peptide having no less than 30% of positively charged residues, as seen from Composition.of.Charge.1 in Table 2. And this can be said with good confidence as the differences between high and low activity class was deemed to be significant according to Welch's $t$-test. Based on the results of this study, the general requirements for high activity HDPs can be verbally summarized as follows: (a) High percentage of positively charged residues, with a correspondingly low percentage of negatively charged residues. The peptide should exhibit a high isoelectric point as well, thus capable of retaining its positive charge over a greater pH range. (b) A low percentage of neutrally charged residues, which possess intermediate solubility in octanol. (c) Cysteine should not be present, methionine residues should be few, but proline residues should be abundant. (d) A high percentage of residues with at least two hydrogen bond donors. (e) A low percentage of residues with either one or no hydrogen bond donor. Meaning that less than two hydrogen bond donors is detrimental for activity. Especially, dipeptides formed of one residue with one hydrogen bond donor and one adjacent residue with no hydrogen bond donor should be avoided. (f) High percentage of residues with high molecular mass and high van der Waals volume. (g) Low percentage of residues buried in hydrophobic core or residues with strand forming inclination. (h) Increased number of dipeptides composed of one helix forming residue and one strand forming residue is beneficial for peptide activity. Note that the notion of high or low is based on the value of a descriptor of high activity class compared to low activity class.

## CONCLUSION

In conclusion, this study represents a systematic exploration of the bioactivity of HDPs via the use of large-scale QSAR modeling where focus was placed on interpretability over performance as to gain more understanding on the antimicrobial activity mechanism. All available results obtained thus far supports the presence of a general antimicrobial mechanism that is independent of specific bacterial strains. This general antimicrobial mechanism is critically dependent on polarity-solubility parameters of the AMP. Previous experimental studies indicated that high positive charge and strong amphipathicity are

the key determinants of AMP antimicrobial activity via membrane lyses. Furthermore, results from this study agrees well with the previous experimental studies in which descriptor patterns indicated that high activity AMPs possessed on average stronger positive charge and could retain this positive charge over a larger pH range. While amphipathicity could not directly be measured by descriptors used in this study, descriptors patterns related to amphipathicity does support the importance of amphipathicity for membrane lyses. Results from this study also indicate that the importance of polarity and solubility permeates the global, local and structural level of AMPs. Results from this study also indicates but does not prove the possibility that while the antimicrobial activity of AMPs is determined by a general mechanism, AMPs require specific configurations to achieve optimal antimicrobial activity against specific bacterial strains.

In terms of the antimicrobial activity mechanism, it should be noted that this study did not discover any new activity influencing factors. Polarity and solubility have been previously known to be important factors influencing antimicrobial activities of HDPs. At first thought, the absence of novel mechanistic discovery might be interpreted as a failure of this study to discover new information on HDPs. As such, an explanation on what new discovery was made is provided hereafter. Chiefly, three new major contributions presented in this study can be succinctly summarized by the following paragraphs. In a nutshell, the main contribution of this study to the field of HDP is that it reaffirms prior knowledge by making use of all available bioactivity data.

1. In terms of dedicated antimicrobial mechanistic investigation, the dataset used in this study is large when compared to existing studies. Prior to the start of this study, an extensive investigation of the field of HDPs was performed by our group as presented in two previous articles (*Li et al., 2016*; *Shoombuatong, Schaduangrat & Nantasenamat, 2018*). In these prior works, it can be attested that prior studies specifically devoted to the antimicrobial strength were almost exclusively experimental in nature, which are necessarily restricted in their sample and descriptor size. Hence, the computational approach of this study allows the exploration of a comparably much larger parameter space.

2. The extensive exploration failed to indicate the presence of new mechanism or mechanisms influencing factors. As such, polarity and solubility are likely all that is necessary for the anti-membrane activities of HDPs. This absence of new mechanism is in itself a novel discovery because its observation is made only possible by the use of the very large descriptor pool in this study coupled with the fact that we had adequately provided interpretative knowledge pertaining to all descriptors used. The large initial descriptor pool coupled with feature selection and descriptor importance calculation allows the observation of as many activity influencing factors as possible, thereby increasing the chance of discovering unknown mechanism. Compared to many related QSARs, the number of descriptors used in this study is very high. A large pool of initial descriptors increases the chance that at least some of them will be strongly correlated with the activity. If such strongly correlated descriptors can be

identified, interpreting their meanings will allow the understanding of activity mechanisms. If a very small pool of descriptors is used, there is a high chance that mechanisms that depend on certain molecular properties cannot be observed because no descriptor measuring those molecular properties were used.

3. Explicit comparisons was made on whether or not different bacterial species/strains required different antimicrobial mechanisms. This study performed a detailed comparison of the mechanisms of different bacterial strains and species of definitive origin (ATCC strains). The conclusion was that in all investigated bacterial strains and species, solubility and polarity are decisive factors governing the antimicrobial activity. Hence, no fundamentally different activity mechanisms can exist amongst the different bacterial strains or species investigated. However, different variations of this general mechanism might be needed for optimal antimicrobial strength.

Hence, this study lends further support to known mechanisms of AMP antimicrobial activity and expands upon previous results. This study also demonstrates that computational modeling in conjunction with extensive human interpretation is capable of yielding readily understandable knowledge while providing the flexibility and efficacy of utilizing prior results and incorporating large number of samples with minimal efforts.

### Funding
This work is supported by the Center of Excellence on Medical Biotechnology (CEMB), S&T Postgraduate Education and Research Development Office (PERDO), Office of Higher Education Commission (OHEC), Thailand. The funders had no role in study design, data collection and analysis, decision to publish, or preparation of the manuscript.

### Grant Disclosures
The following grant information was disclosed by the authors:
Center of Excellence on Medical Biotechnology (CEMB), S&T Postgraduate Education and Research Development Office (PERDO), Office of Higher Education Commission (OHEC), Thailand.

### Competing Interests
The authors declare that they have no competing interests.

### Author Contributions
- Hao Li conceived and designed the experiments, performed the experiments, analyzed the data, prepared figures and/or tables, authored or reviewed drafts of the paper, approved the final draft.
- Chanin Nantasenamat conceived and designed the experiments, analyzed the data, contributed reagents/materials/analysis tools, prepared figures and/or tables, authored or reviewed drafts of the paper, approved the final draft.

## Data Availability

Data is available at GitHub: https://github.com/chaninlab/antimicrobial-peptide-QSAR/.

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
