# Peer review of "Toward insights on determining factors for high activity in antimicrobial peptides via machine learning"

_PeerJ, doi:10.7717/peerj.8265_

## Round 0.1 · original submission · Major Revisions

Both reviewers have comments about the algorithm you are using. Maybe you should consider making your programs/ R scripts available to the community, e.g. using Github.

Additionally, the grammar/ writing of your manuscript should be proofread/ corrected by a native speaker/ editing service.

Reviewer 1 ·

Basic reporting

This work has multiple grammatical and orthographical errors as well as an ambiguous use of the English language which makes it difficult to read through. It also lacks several relevant references in AMPs/HDPs as discussed below.

1a. Ambiguous sections

Line 15: the review by Gaspar et al. refers to AMPs with anticancer properties, not HDPs in general. In that regard, the authors should refer any work from Robert E.W. Hancock such as https://www.frontiersin.org/articles/10.3389/fchem.2019.00043/full or https://www.ncbi.nlm.nih.gov/pubmed/22173434

Lines 16-19: All this information can be found in the aforementioned reference. The authors added other specific references such as Salger et al. (2016) for anti-protozoans or Kalenik et al. (2016) for antiviral HDPs, that we believe are unnecessary. Moreover, they cited their own reference Li et al. (2016) that refers to the effects of D-amino acids on HDPs bioactivity. Interestingly, that particular reference is cited several times across the manuscript in an unjustified manner.

Lines 53-54: The authors mentioned that “this study is not merely aimed at constructing prediction models of HDP antimicrobial activities but this study interprets the QSAR model on the basis of prior knowledge in the field of HDPs…” however in their methodologies, they assembled a dataset of 1641 peptide sequences divided into 3 levels/classes (high, medium, low) for which they calculated 760 molecular descriptors. However, following multivariate analysis and predictive modeling, the authors built a random forest-based classifier as described lines 208-209 “In the construction of the QSAR model, descriptors were used as the input data matrix while the assigned activity levels for each of the AMP was used as the expected output vector”. Reading further the manuscript, it becomes clear that the authors were not trying to produce the best predictive model but the most interpretable one (line 193). If we take that argument in consideration, why not exploring other tree-based models such as the more interpretable decision tree or gradient boosting.

Lines 62-64: “While numerous online databases on HDPs already exists, the DBAASP database appears to be the only one that provides an extensive repository of bioactivity and target information.” It will be appropriate to mention with one or more references these online databases the authors are referring to.

Lines 120-129: In this paragraph, the authors explained badly that they will use a multi-level classifier instead of regression models by binning the MIC activity data (in M) into 3 levels: high, intermediate and low. The M cut-offs they used to form their levels have been omitted and they should be mentioned in this section. Interestingly, they have not used the word ‘classifier’ or ‘classification model’ in this manuscript, we strongly encourage them to do so for the sake of clarity.

Line 132: Please explain in more details why/how a single AMP may possess significantly different MIC values for each strain.

Lines 133-153: The authors explained at great length that several peptide sequences were reported multiples times (up to 5 times) with different MIC values. Most peptides were only reported once. All MIC values from each unique sequence were not averaged as they “landed” within the same level of MIC activity, therefore each unique peptide will be administrated one activity level. In general, activity binning and removing replicates are inherent parts of data pre-processing. That paragraph should be condensed.

1b. Orthographic and grammatical errors:

Lines 12-13: poor English usage
Line 16: amphipathicity
Line 20: has
Line 25: interact
Lines 24-27: poor English usage
Lines 29-46: Re-phrase the two paragraphs to better highlight the strengths and weaknesses of applying computational techniques to HDPs/AMPs
Line 36: had
Line 40: HDP-related topics
Line 53: aiming
Line 60: all peptide sequences
Line 62: exist
Line 68: peptide and bioactivity data that are
Line 75: deduced
Line 83: the bacterial strain names as listed above are not …
Lines 92 and 109: MIC > 50 M
Line 202: Kuhn Kuhn
Lines 330-333: poor English usage
Line 335: throughout
Lines 337-339: poor English usage
Line 354: an
Line 368: “And as shall be elucidated”? Please re-phrase.
Line 375: Recall that… are the authors talking to the reader directly?
Line 384: For
Line 398: “As can be seen” ? Please re-phrase.
Line 407: …PROFEAT which was originally developed…
Line 515; “And the observation made here…” Please re-phrase.
Table 2 caption: p-value < 0.05

The raw data were shared on Github.

Lots of speculation or hypotheses with limited evidence.

Experimental design

The present manuscript describes the application of random forest (RF) classification for predicting the antimicrobial activity of HDPs and understands which peptide molecular descriptor(s) would impact on the level of antimicrobial (MIC) activity (high, medium, low). The original model itself uses a dataset of 1641 peptides (observations) described by 760 molecular descriptors (variables). The class variable displays 3 levels of the MIC (minimum inhibitory concentration) activity data (in uM): high, medium/intermediate and low.

The scope of this study is within the Aims and scope of the journal PeerJ in the area of Biological and Health sciences.

In itself, this study is an interesting piece of work, it executes and discusses most parts of building a QSAR (machine learning) model: data collection, data preprocessing, exploratory data analysis e.g. multivariate analysis/supervised feature selection CfsSubsetEval, data splitting and 10-fold cross-validation, model building (500-1500 trees) and minimal model optimization. Did the authors normalize/standardize their data prior to execute any multivariate analysis and predictive modeling? It was not mentioned.

Interestingly, the authors did not provide the R script on Github where the multivariate analysis and random-forest classifiers were executed.

However, by line 231, the manuscript became confusing. The authors created two different RF-based classification models where the output variable (class) is either 2 activity levels (high/low) for each of the 7 strains or 3 activity levels (high, low, inactive) based on another set of active AMPs /inactive control UniProt peptides. These UniProt control peptides should be mentioned in the “Data set” section. What happened to the original 3-class random forest classification (high, medium/intermediate and low)? Instead, this study focused on building “a predictive model for each strain using a large collection of different peptides (line 352)”.

Validity of the findings

The authors found that descriptors of different classes (local sequence, global, structural) participated to predict the antimicrobial activity of the peptides in a strain-specific manner. Figure 2 perfectly illustrates that point where the descriptors are ranked by importance per strain. While the descriptors differ in terms, the authors concluded that the important ones were all describing the same property; polarity-solubility as mentioned in lines 409 to 412: “The important composition descriptors were parameters for charge, ClogP, hydrophobicity, side chain hydrogen bond donor, van der Waals volume, polarity, polarizability, solvent accessibility, surface tension, and secondary structure. With the exception to secondary structure, all the composition descriptors were parameters closely related to peptide charge and solubility in a vice versa manner.” They also concluded that these trends were not strain-specific. The authors could sum up the importance per strain of the descriptors (either local sequence, global or structural) that refer to the same property: polarity/charge or solubility (as shown in Figure 2) to see if in general this property is equally represented among the different strains.

In the main body of the manuscript (lines 390-, the authors discussed at length the size and the impact (Welch’s t-test, null hypothesis) of all global, local sequence and structural descriptors. They illustrated their discussion with a massive Table 2. They spend enormous efforts to describe the impact of individual descriptor which makes the reader (the medicinal chemist) forget the big picture. It is good to remind the authors that such computational study aims at providing guidelines for the lab scientists who pay little attention into individual descriptors to design (antimicrobial) peptides.

While this study has the merit to apply QSAR modelling between the molecular descriptors of 1641 peptides and their antimicrobial activity, their findings are not completely new “this study did not discover any new activity influencing factors”. Instead the authors should suggest a series of guidelines to support/debunk the design of antimicrobial membranolytic peptides - based on their findings (polarity and solubility). At best, this computational study should be provided with experimental results.

Reviewer 2 ·

Basic reporting

When cited most of the references, a parenthesis is missed, for instance, in line 12 one can read “and has been extensively reviewed before Hiltunen et al. (2017) and” when it should be “and has been extensively reviewed before (Hiltunen et al. (2017)) and”
The same happens in many places and makes the reading uncomfortable
Some examples of grammar mistakes are:
1. lines 256-257 “Support the claim that the random forest models build in this study is able to differentiate the activity” versus “Support the claim that the random forest models build in this study ARE able to differentiate the activity”
2. line 407 “as done by PROFEAT was original developed by Dubchak” versus “as done by PROFEAT was ORIGINALLY developed by Dubchak”
3. line 454 “A remarkable consistency can observed when” versus “A remarkable consistency can BE observed when”
4. line 467 “any descriptors” versus “any descriptor”
5. line 654 “lyses can occurred” by “lyses can occur”
6. line 660 “Results from this study also indicates” versus “Results from this study also indicate”
7. line 699 “we had adequately provide” versus “we had adequately provided”
8. line 704 “correlated with the activity” versus “correlate with the activity”
9. line 707 “measuring those molecular property were used” versus “measuring those molecular properties were used”
Some typos like:
1. line 379 “it would indicate that that descriptor” versus “it would indicate THAT descriptor”
2. line 506 “(¡1.08)”
3. Table 2, caption, “was p-value ¡0.05.”
4. line 609 “few chain segements” versus “few chain segments”
5. line 664 “obtained thusfar supports” versus “obtained thus far supports”
A brief but clear explanation, besides what the authors provide in lines 214-222, explaining the CfsSubsetEval algorithm is required, this is a central tool in your analysis, the reader should understand from your work how this algorithm decides which set of features is relevant.
The explanation for Structural descriptors in lines 266-268 is insufficient, you need to provide more details.

Experimental design

In Table 1 how can you explain having a higher performance for 3 activity levels than for 2? Please extend your explanation in line 251.

A more profound analysis of the data sets is required, what is the sequence diversity of each set, what is the sequence similarity of sequences between different strains?
Within the same strain:
1) what is the sequence similarity between highly active sequences?
2) what is the sequence similarity between low active sequences?
3) what is the sequence similarity between high and low active sequences?
4) what about the same analysis of 1-3 but now in the descriptor space?

How can you include the differences found in this analysis to normalize the results shown in Figure 1?

A PCA on the set of data per strain, labeling (with color, shape, etc) the projected dots with their activity levels may produce useful information.

Validity of the findings

As the authors mention (lines 358-360), “That is, if peptides used for building the activity prediction models for a strain is changed, the list of important descriptors and their importance values will also change as well.”
In the context of this statement, how can you be confident of your findings?, answering the questions listed in the Experimental Design section may provide you with information about, for instance, for which strain the descriptors are more informative because you have already the amount and diversity of peptides needed to state statistical significant conclusions.

Additional comments

The authors present interesting ideas, understanding the importance of descriptors associated with a given activity is of utmost relevance. Trying to provide the rationale to the decision taken by ML models and finding biologically relevant explanations is an active area of research. I see this work promising although still in an early stage of development. Technical aspects such as data analysis should be improved a great deal.

---

## Round 0.2 · Minor Revisions

Both reviewers agree that your paper improved substantially. However, a minor revision would further improve the quality of your manuscript.

Reviewer 1 ·

Basic reporting

Most comments from the first review have been taken into consideration – the use of the English language has been improved. Numerous relevant references and paragraphs have been added to the original manuscript which makes the document much more understandable and readable. Overall, the manuscript has a professional structure including figures and tables. All speculations have been removed.

Experimental design

The present manuscript is in the Scope of the journal, it describes the application of a random forest (RF) classification for predicting the antimicrobial activity of HDPs/AMPs and understands which peptide molecular descriptor(s) would affect their level of antimicrobial (MIC) activity “The core objective of this study is to provide a readily understandable interpretation of relationship of 287 peptide molecular descriptor patterns and antimicrobial activity”. The authors explored the MIC (minimum inhibitory concentration) activity data in uM across 7 bacteria strains (a) Bacillus subtilis ATCC 6633, (b) Enterococcus faecalis ATCC 29212, (c) Staphylococcus aureus ATCC 6538, (d) Staphylococcus aureus ATCC 25923, (e) Escherichia coli ATCC 25726, (f) Escherichia coli ATCC 25922, (g) Pseudomonas aeruginosa ATCC 27853. The different MIC activity values were binned into 3 levels: high, medium and low. Two classification models have been developed – one tertiary model used a dataset of 1460 peptides divided into the aforementioned 3 classes (high, medium, low). A second binary classification model used 972 peptides where the medium class has been excluded. Each peptide was described by 760 molecular descriptors as follows: (a) 2 parameters pertaining to the molecular property namely the molecular weight (MW) and isoelectric point (PI), (b) 20 amino acid composition descriptors, (c) 400 dipeptide composition descriptors, (d) 2 sequence coupling number as measured by Schneider-Werder and Grantham distance, respectively, (e) 42 Quasi-sequence order descriptors as measured by Schneider-Werder and Grantham distance, respectively, (f) 42 composition, 42 transition and 210 distribution descriptors of 14 properties [(1) hydrophobicity, (2) van der Waals volume, (3) polarity, (4) polarizability, (5) charge, (6) secondary structure, (7) solvent accessibility, (8) surface tension, (9) molecular weight, (10) solubility in water, (11) number of hydrogen bond donor in side chain, (12) number of hydrogen bond acceptor in side chain, (13) ClogP, (14) amino acid flexibility index]. Of these descriptors or features, the authors proceeded to the selection process CfsSubsetEval and they identified 30 global, 60 local sequence and 88 structural descriptors as important descriptors for their models. Of these 178 descriptors, 138 were identified as unique descriptors. In itself, this study discusses most parts of building a QSAR model: data collection, data preprocessing, exploratory data analysis e.g. multivariate analysis/supervised feature selection CfsSubsetEval, data splitting and 10-fold cross-validation, model building (500-1500 trees) and minimal model optimization.

Validity of the findings

The authors studied the role of descriptors of different classes (local sequence, global, structural) to predict the antimicrobial activity of the peptides in a strain-specific manner. For example, “the global descriptor patterns agree very well with what is known about the HDP antimicrobial activity mechanism and does not indicate strain-specific activity mechanism.” Despite their efforts, they could not find strain-specific descriptors. Instead, they observed that most descriptors were related to the polarity-solubility property and there was a general dependence of antimicrobial activity on these descriptors and no strain specificity.

Additional comments

This study has the merit to apply QSAR modelling between the molecular descriptors derived from peptide sequences to predict antimicrobial activity (MIC). While their findings are not completely new, their overall analysis and conclusions are satisfactory and worth publishing.

Annotated reviews are not available for download in order to protect the identity of reviewers who chose to remain anonymous.

Reviewer 2 ·

Basic reporting

The paper has improved a great deal.

Experimental design

See general comments

Validity of the findings

See general comments

Additional comments

This reviewer is satisfied with all but one of the answers provided by the authors that I would like to address:

My original comment was:
A more profound analysis of the data sets is required, what is the sequence diversity of each set, what is the sequence similarity of sequences between different strains?
Within the same strain:
1) what is the sequence similarity between highly active sequences?
2) what is the sequence similarity between low active sequences?
3) what is the sequence similarity between high and low active sequences?
4) what about the same analysis of 1-3 but now in the descriptor space?
You can forget about point 4. The authors' argument for not dealing with the suggested analysis can be simplified into two points:
1) The time cost and length of the analysis
2) The aim of the paper’s analysis is not in the sequence space.
The time required to perform analysis from 1 to 3 is little, you can use any software for computing sequence similarity. Regarding the length, it can be solved by adding the results as a supplementary file.
I clearly understand the point the authors try to make about the descriptor space, however, we should not forget that we do analysis in the descriptors space because it will be more informative, simpler, and mainly because the sequence space is not enough to discriminate the desired properties.
My point of doing the data analysis has to do with the following reasoning: what happens if you can do better by just measuring the distance in the sequence space?
For instance, assume that, in the sequence space, any two sequences of class A are closer to each other than any pair of sequences one of class A and another of class B, and that the same is true for any pair of sequences in class B. Then the sequence similarity will be enough for separating the classes.
This situation is probably not happening in your case, in your data set perhaps there exist at least a pair of sequences of the same class that are farther away from each other than some pair of sequences from different classes. The experiments I propose will help you to show that this last situation is actually your case and that you need your descriptor’s space approach. However, any other way you have to show that the descriptors approach is needed (i.e., that you will have a problem discriminating at the sequence level) will be sufficient.

---

## Round 0.3 · accepted · Accept

The minor comments on the previous version were addressed adequately.